



# Asthenospheric anelasticity effects on ocean tide loading in the East China Sea region observed with GPS

Junjie Wang[1*,2], Nigel T. Penna[2], Peter J. Clarke[2], Machiel S. Bos[3]

[1] School of Earth Sciences and Engineering, Hohai University, Nanjing, China
[2] School of Engineering, Newcastle University, Newcastle upon Tyne, UK
[3] SEGAL, University of Beira Interior, Covilhã, Portugal
[*] Corresponding author, email wangjunjie_gnss@qq.com

## Abstract

Anelasticity may decrease the shear modulus of the asthenosphere by 8-10% at semi-diurnal tidal periods compared with the reference 1 s period of seismological Earth models. We show that such anelastic effects are likely to be significant for ocean tide loading displacement at the $M_2$ tidal period around the East China Sea. By comparison with tide gauge observations, we establish that NAO99Jb is the most accurate numerical ocean tide model in this region, and that related errors in the predicted
$M_2$ vertical ocean tide loading displacements will be 0.2-0.5 mm. In contrast, GPS observations on the Ryukyu Islands (Japan), with uncertainty 0.2-0.3 mm, show discrepancies of over 1.5 mm with respect to ocean tide loading displacements predicted using the purely elastic radial Preliminary Reference Earth Model. We show that the use of an anelastic PREM-based Earth model reduces these discrepancies to no more than 0.8 mm, which is of the same order as the sum of the remaining errors
due to uncertainties in the ocean tide model and the GPS observations. Use of a regional Earth model based on the laterally-varying S362ANI, with or without further empirical tuning, results in minor additional improvements in fit.

## 1 Introduction

The periodic redistribution of ocean mass around the Earth's surface due to ocean tides deforms the solid Earth, a phenomenon known as ocean tide loading (OTL). The resulting OTL displacements can reach several centimetres in the vertical component and more than one centimetre in the horizontal components, with the Earth's response to the OTL depending strongly on the material properties within its interior (Farrell, 1972). In the past two decades, Global Positioning System (GPS) data
analysis techniques have been developed to directly measure OTL displacements with millimetre accuracy, and even sub-millimetre accuracy at some frequencies (e.g., Allinson et al., 2004; Thomas et al., 2007; Yuan et al., 2009; Penna et al., 2015). With parallel substantial advancements in the accuracy of global ocean tide models (Stammer et al, 2014; Ray et al., 2019), comparisons of GPS-observed and predicted (modelled) OTL displacements have several times revealed the deficiencies
of using spherically symmetric, non-rotating, elastic and isotropic (SNREI) Earth models. One of the reasons for these deficiencies is that these models have been derived from seismic data and represent the Earth's elastic properties at a reference period of 1 s, but have typically been assumed to be directly applicable at tidal frequencies.





Ito et al. (2009) analysed the average amplitude ratio between GPS tidal displacement observations and an Earth tidal model (including OTL and Earth body tide) across Japan, finding that the positive trend of amplitude agreed with predictions from inelastic Earth models. Ito and Simons (2011) further attempted to invert GPS-observed displacements for one-dimensional profiles of the elastic moduli
and density beneath the western United States, demonstrating the limitations of the Preliminary Reference Earth Model (PREM) (Dziewonski and Anderson, 1981). Also, Yuan and Chao (2012) and Yuan et al. (2013) reported continental-scale spatially coherent differences between GPS-observed and predicted OTL displacements at sites located more than 150 km inland from the coastline, and attributed these differences to elastic and inelastic deficiencies in the a priori Earth body tide model.
More recently, Bos et al. (2015) showed for western Europe that large discrepancies exist between GPS-observed and modelled OTL displacements, arising from disregarding anelastic dispersion in the asthenosphere that occurs when the elastic constants of the Earth model are modified to be applicable at tidal periods. Such an effect could bring about a reduction of around 8-10% of the shear modulus in the asthenosphere at tidal frequencies. In addition, Martens et al. (2016) observed spatial
coherence among residual $M_2$ OTL displacements across South America, postulating deficiencies in the a priori SNREI Earth models.

Bos et al. (2015) showed the feasibility of representing the behaviour of the asthenosphere across an absorption band from seismic to tidal frequencies by a constant quality factor $Q$, which provides a
rough transformation to account for the anelastic dispersion effect. Hence, it can be postulated that the asthenosphere should always produce ~8.5% OTL displacement discrepancies with respect to a purely elastic PREM-based Earth model, not only in western Europe where Bos et al. demonstrated this effect, but all over the world. However, these discrepancies will not be equally observable in all localities, either because ocean tide amplitudes are too small within the 50-250 km distance range
from the analysis point that samples asthenospheric behaviour, or because regional uncertainties in ocean tide models are too large to be able to attribute any observed discrepancy to the Earth model. To identify regions where the findings of Bos et al. (2015) are testable, we have examined the global distribution of a 'detectability ratio'. This is defined as the ratio between the elastic-anelastic OTL displacement discrepancy (taken to be the difference between OTL predicted using a purely elastic
PREM Green's function, as described in Section 3, and that using Bos et al.'s anelastic S362ANI($M_2$) Green's function) as the numerator, and the combination of expected GPS observational and ocean tide model related errors as the denominator. For the latter, the ocean tide model related error is characterised as the standard deviation (STD) of the predicted elastic OTL displacements at each location, using each of the DTU10, EOT11a, FES2014b, GOT4.10c, HAMTIDE11a, NAO99b,
OSU12, and TPXO9-Atlas numerical ocean tide models (see Table 1 for references), and the GPS observational error is assigned a STD of 0.3 mm following Penna et al. (2015).

Figure 1a shows a global 1/8° grid of detectability ratio for the $M_2$ vertical OTL displacement, which is unfavourable (less than one) for most inland and deep ocean regions. Many of the areas where it
exceeds one, such as off the coasts of southern Greenland, eastern Africa and central America, are poorly sampled with continuously-operating GPS networks. However, the East China Sea (ECS) region exhibits a favourable combination of large OTL displacements and fairly consistent ocean tide models across much of it, so the detectability ratio here exceeds three across a wide area, and contains a healthy distribution of long-running GPS sites (Figure 1b shows the 102 GPS sites used).
Accordingly, we have selected this as a suitable region for an independent test of Bos et al.'s (2015)





conclusions. A further attraction of this region for the testing of Earth models is that its position overlying a subduction zone means that it represents a very different tectonic setting to the mature passive margin in western Europe studied by Bos et al.

Figure 1c shows the predicted $M_2$ vertical OTL displacements across the ECS region using the FES2014b ocean tide model (Carrère et al., 2016) and an elastic PREM Green's function. It can been seen that the $M_2$ vertical OTL displacement amplitudes are as large as 20-25 mm around the Ryukyu Islands and on the southeast coast of China, so the anelastic OTL displacement discrepancies would be expected to be about 2 mm and therefore detectable using GPS. Overall, the accuracy of recent
ocean tide models is believed to be good, e.g. Stammer et al. (2014) show sub-centimetre $M_2$ root mean square (RMS) agreement between bottom pressure observations and seven recent models in the deep oceans globally and additionally, the FES2014b model has been suggested as providing a clear advancement in global ocean tide modelling (Ray et al., 2019). However, the fact that the tides in the ECS are large and complex owing to the irregular geometry of the basin (Lefèvre et al., 2000) implies
that careful evaluation of the ocean tide models is still necessary in this region to ascertain the optimal model, and thus minimise the effect of errors in ocean tide models on the OTL predictions.

In this paper, we first assess the accuracy of a selection of up-to-date ocean tide models in the ECS, and quantify their contribution to the predicted OTL error budget. We then describe the kinematic
GPS analysis approach for obtaining the observed OTL displacements. Finally, we examine the evidence of asthenospheric anelasticity effects in the ECS region based on the GPS-observed OTL displacements. We consider the $M_2$ constituent and the vertical component of OTL displacement, as these are dominant in the ECS region.

**Figure 1 (a)** Global distribution (1/8° grid) of $M_2$ 'detectability ratio' of difference between vertical OTL displacements predicted using purely elastic and anelastic Green's functions to uncertainty in residual OTL displacements predicted using eight ocean tide models and the GPS observational error. **(b)** Detectability ratio in the East China Sea (ECS) region, showing as triangles the GPS sites used in this study. **(c)** The $M_2$ vertical OTL displacement amplitudes and Greenwich phase lags for a 1/8° grid across the ECS region using the FES2014b ocean tide model and an elastic PREM Green's function.





## 2 Ocean tide model accuracy assessment using tide gauges

A pre-requisite for using GPS measurements of OTL displacement for evaluating the Earth's interior material properties is that the impact of ocean tide model errors on the predicted OTL displacement is understood and found to be near negligible. Therefore, we first evaluate the quality of ocean tide
models in the ECS region (considered throughout this paper as 116° to 133° east in longitude and 23° to 42° north in latitude) by assessing their consistency with each other and by comparing them with tide gauge observations.

To date, no single ocean tide model has been demonstrated as optimal in all regions of the world
(Stammer et al., 2014; Ray et al., 2019), so we selected eight recent global (DTU10, EOT11a, FES2014b, GOT4.10c, HAMTIDE11a, NAO99b, OSU12, TPXO9-Atlas) models and one regional (NAO99Jb) model for the quality assessment. The key features of the models are listed in Table 1. All models, except for GOT4.10c, directly assimilate TOPEX/Poseidon (T/P) altimeter data plus, for some of the models, data from one or more of the ERS-1/2, Geosat Follow-on (GFO), Jason-1/2,
Envisat and ICESat altimetry satellites, as well as tide gauge data. FES2014b, HAMTIDE11a, NAO99b and TPXO9 are barotropic data-assimilative models. DTU10 and EOT11a are both based on an empirical correction to the global hydrodynamic tide model FES2004 (Lyard et al., 2006), while the a priori model for GOT4.10c is a collection of global and regional models blended at mutual boundaries. OSU12 is a purely empirical model determined by analysis of multi-mission satellite
altimeter measurements. TPXO9-Atlas is obtained by combining the base global TPXO9 and local solutions for all coastal areas including around Antarctica and the Arctic Ocean. The regional model, NAO99Jb, covers the area from 110° to 165° east in longitude and from 20° to 65° north in latitude, including the whole area of the ECS, and assimilates more local tide gauge data than do the other models.


**Table 1** Summary of the selected ocean tide models.

| Model | Data assimilated[a] | Resolution | Type[b] | Author / Reference |
|---|---|---|---|---|
| DTU10 | T/P, ERS-2, GFO, Jason-1/2, Envisat | 1/8° | E | Cheng and Andersen (2011) |
| EOT11a | T/P, ERS-2, Jason-1/2, Envisat | 1/8° | E | Savcenko and Bosch (2012) |
| FES2014b | T/P, ERS-1/2, Jason-1/2, Envisat, TG | 1/16° | H | Carrère et al. (2016) |
| GOT4.10c | ERS-1/2, GFO, Jason-1/2, ICESat | 1/2° | E | Ray (2013) |
| HAMTIDE11a | T/P, Jason-1 | 1/8° | H | Taguchi et al. (2014) |
| NAO99b | T/P | 1/2° | H | Matsumoto et al. (2000) |
| NAO99Jb | T/P, TG | 1/12° | H | Matsumoto et al. (2000) |
| OSU12 | T/P, GFO, Jason-1, Envisat | 1/4° | E | Fok (2012) |
| TPXO9-Atlas | T/P, ERS-1/2, Jason-1/2, Envisat, TG | 1/30° | H | Egbert and Erofeeva (2002) |

[a]T/P, TOPEX/Poseidon; GFO, Geosat Follow-on; TG, tide gauge.

[b]E, empirical adjustment to an adopted a priori model; H, assimilation into a barotropic hydrodynamic model.

To evaluate the consistency among the different ocean tide models for the dominant $M_2$ constituent, all models were bilinearly interpolated on to a common 1/16° grid across the ECS region and the STDs of the phasor differences from the mean were computed per grid point, as shown in Figure 2. It can be seen that away from the coastlines, all models are quite similar with the STD no more than 1 cm, which likely arises because they have more or less assimilated the same altimeter data, albeit





over different durations. However, closer to the coast large inter-model discrepancies arise, especially in the Seto Inland Sea, and near the coast of eastern China and western Korea, where the STD exceeds 30 cm in places.

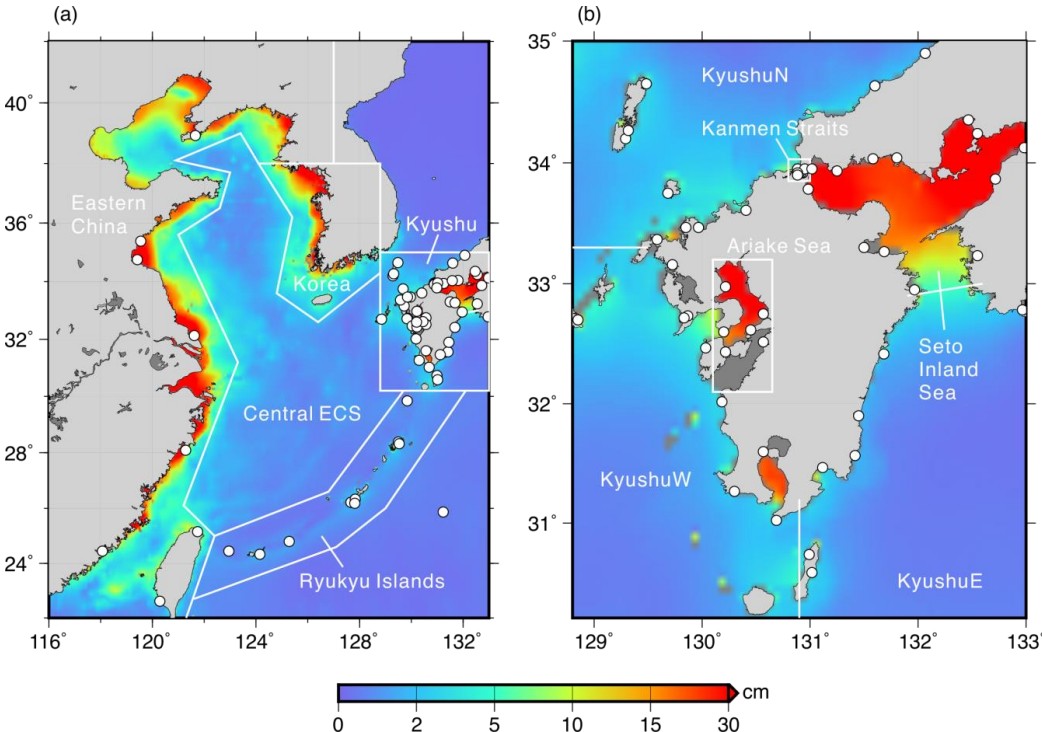

**Figure 2** The $M_2$ standard deviations for nine ocean tide models (DTU10, EOT11a, FES2014b, GOT4.10c, HAMTIDE11a, NAO99Jb, NAO99b, OSU12 and TPXO9-Atlas). **(a)** shows the whole East China Sea (ECS) region, while **(b)** is an enlargement of the Kyushu sub-area of (a). The white labelled polygons define the sub-areas for which the quality of the ocean tide models has been evaluated, and the white dots represent the locations of coastal tide gauges.

To ascertain which models are the cause of the large STDs in some sub-areas, and to assess their accuracy, we compared each model with observations from 75 coastal tide gauges (58 from the Japan Oceanographic Data Centre and 17 from the University of Hawaii Sea Level Center) in the ECS region, as shown in Figure 2. Using the UTide package (Codiga, 2011), the tidal constants observed at these locations were deduced from hourly sea level time series spanning 4 to 69 years, with a median time-series length of 26 years. For time series shorter than 18.6 years, we applied nodal corrections at exact times during the harmonic tidal analysis (Foreman et al., 2009), instead of regarding them as constant values.

In order to investigate in detail the problematic areas of eastern China, western Korea and the Seto Inland Sea, the region is divided into the separate sub-areas shown in Figure 2, basically in accordance with the zones of inter-model discrepancy. Moreover, for the sake of describing the ocean tide model errors as precisely as possible in the next section, the sub-area denoted as Kyushu is further divided. The $M_2$ phasor difference between each model and each tide gauge was computed, and the RMS of





these differences per model for all tide gauges in each sub-area are listed in Table 2.

For eastern China, FES2014b and NAO99Jb perform quite well (RMS of 10-12 cm), whereas DTU10 and EOT11a are the worst models (RMS of 47-59 cm). This could be explained by the fact that the
FES2004 model, on which DTU10 and EOT11a are both based, has several grossly incorrect tidal values in this area owing to insufficient satellite altimetry data available at the time. RMS agreements of better than 4 cm between tide gauge observations and each of the models are obtained for the Ryukyu Islands sub-area, except for TPXO9-Atlas. This is despite TPXO9-Atlas having the finest resolution among the models of 1/30°, whereas the coarser (1/2°) GOT4.10c and NAO99b models
have better than 4 cm RMS agreement. Around the island of Kyushu, the observations compare consistently well with FES2014b and NAO99Jb (RMS lower than 4 cm), while the comparisons are poor for DTU10, EOT11a, HAMTIDE11a, OSU12 and TPXO9-Atlas along the west coast of Kyushu, and for GOT4.10c and NAO99b along the north coast of Kyushu. NAO99Jb exhibits the best agreement with the observations in the Ariake Sea and Seto Inland Sea, which is expected as it
assimilates data from 219 local tide gauges (Matsumoto et al., 2000). This also results in NAO99Jb being more accurate than NAO99b in most parts of the ECS region. However, the agreement between NAO99Jb and the tide gauges is no better than the other models in the Kanmen Straits, because the tide gauges there were installed in 2011, after the release of NAO99Jb, and hence none of their data have been assimilated. Nonetheless, NAO99Jb is the most accurate ocean tide model in the ECS
region as a whole.

**Table 2** The root mean square (in cm) of the $M_2$ phasor differences between each of the nine ocean tide models and the tide gauge observations in each defined sub-area of the East China Sea region.

| Area | DTU10 | EOT11a | FES2014b | GOT4.10c | HAMTIDE11a | NAO99Jb | NAO99b | OSU12 | TPXO9-Atlas |
|---|---|---|---|---|---|---|---|---|---|
| Eastern China | 47.4 | 59.1 | 9.6 | 30.1 | 42.5 | 11.7 | 35.4 | 18.3 | 34.5 |
| Ryukyu Islands | 3.1 | 3.2 | 3.1 | 3.9 | 3.4 | 2.4 | 3.9 | 3.6 | 11.0 |
| KyushuW | 15.3 | 19.0 | 3.3 | 6.6 | 17.5 | 3.7 | 6.8 | 13.8 | 8.1 |
| Ariake Sea | 29.6 | 29.1 | 29.2 | 46.5 | 34.8 | 3.1 | 34.8 | 39.6 | 23.7 |
| KyushuE | 3.6 | 3.7 | 2.5 | 4.8 | 4.0 | 3.0 | 5.6 | 3.9 | 4.6 |
| KyushuN | 2.9 | 3.0 | 1.8 | 8.2 | 2.7 | 2.1 | 7.3 | 5.8 | 6.6 |
| Seto Inland Sea | 34.4 | 43.0 | 31.3 | 42.1 | 57.3 | 3.3 | 46.3 | 36.3 | 38.0 |
| Kanmen Straits | 15.6 | 17.6 | 14.5 | 12.9 | 16.8 | 16.2 | 16.8 | 11.8 | 11.9 |

**3 Impact of ocean tide model errors on OTL displacement**

In this section we assess the impact of ocean tide model errors on the predicted OTL displacements, which is needed to ensure the confident geophysical interpretation of the GPS-observed OTL displacement residuals considered thereafter. For a particular tidal constituent, the OTL displacement $u$ at a point $\boldsymbol{r}$ on the Earth's surface may be computed (predicted) with the following convolution
integral (Farrell, 1972):

$$u(\boldsymbol{r}) = \int_{\Omega} \rho G(|\boldsymbol{r} - \boldsymbol{r}'|) Z(\boldsymbol{r}') d\Omega \tag{1}$$

where $\Omega$ represents the global water areas, $\rho$ is the density of seawater, $G$ is a Green's function that





describes the displacement at $r$ from a unit point load, and $Z$ is the tide height at $r'$, written as a complex number to include both the amplitude and varying phase-lag. Here, the convolution integral is determined by numerical integration, and may be written as:

$$u(\boldsymbol{r}) = \sum_{\Omega} \rho Z_i G_i \qquad (2)$$

where $G_i$ here is the integrated Green's function for the $i$th element of $\Omega$, as per Agnew (1997), and the tidal heights $Z_i$ are represented over $\Omega$ by inputting a global ocean tide model.

Bos et al. (2015) took the STD of predicted OTL displacements computed per point for a set of ocean tide models as the error contribution of the ocean tide models in western Europe, assuming that there
were no systematic biases shared by the models. However, we have shown in Section 2 that for the ECS region, the STD among the models is not always a good indicator of their accuracy. To check this, $M_2$ vertical OTL displacements were computed for a 1/8° grid across the ECS region for each of the nine ocean tide models (NAO99Jb was augmented globally outside its boundary by FES2014b) using the SPOTL (NLOADF) software version 3.3.0.2 (Agnew, 1997). A Green's function computed
based on the isotropic, purely elastic version of PREM was input (as for all elastic PREM-generated results in this paper) and is provided in Appendix A. As the GPS sites considered in this study are on land, the upper 3 km water layer in PREM was replaced with the density and elastic properties from the underlying rock layer. The OTL displacement STDs among the models per point are shown in Figure 3a, and it can be seen that the distribution of the STDs is similar to those shown for the ocean
tide models in Figure 2, with large STDs of up to 2.5 mm arising around eastern China, western Korea and the Seto Inland Sea. However, as shown in Section 2, these large STDs arise from large errors in some (but not all) of the nine ocean tide models and NAO99Jb was shown to be the most accurate model across the ECS region. Therefore, it is unreasonable to use the inter-model STD as an indicator of OTL displacement accuracy for the ECS region. Instead, we now present an approach which allows
us to quantify (to first order) the resulting OTL displacement prediction error individually for a particular ocean tide model.

Assuming the ocean is divided into $k$ specified water areas $\Omega_k$ (e.g., as per Table 2), and that the ocean tide model error magnitude per area is $\delta_k$, the corresponding OTL accuracy $\delta u_k$ is:

$$\delta u_k(\boldsymbol{r}) = \sum_{\Omega_k} \rho \delta_k G_i \qquad (3)$$

Then, assuming no correlation between each of the $k$ areas, the total OTL displacement prediction error may be computed as:

$$\delta u(\boldsymbol{r}) = \sqrt{\sum \delta u_k^2(\boldsymbol{r})} \qquad (4)$$

To evaluate the OTL error using equation 4 for NAO99Jb, the most accurate ocean tide model in the ECS region, we define the ocean tide model errors for the separate sub-areas (as per Figure 2) as the RMS difference between NAO99Jb and the tide gauge observations within the sub-area (Table 2). For the Korea sub-area, although no tide gauge data source is available, the error of NAO99Jb for Korea can be estimated as the mean value of the RMS of the areas around Kyushu excluding the
Kanmen Straits, considering the fact that NAO99Jb also assimilated the tide gauge data around Korea.





The 'other water areas' (comprising the central ECS sub-area and all other global water areas not named in Figure 2) are either open oceans, or narrow coastal areas that are far from the ECS. To be conservative, a slightly larger value of 0.7 cm is chosen as the RMS error of NAO99Jb and its complement of FES2014b for these areas, according to the largest RMS model differences of 0.66 cm for deep oceans inferred by Stammer et al. (2014).

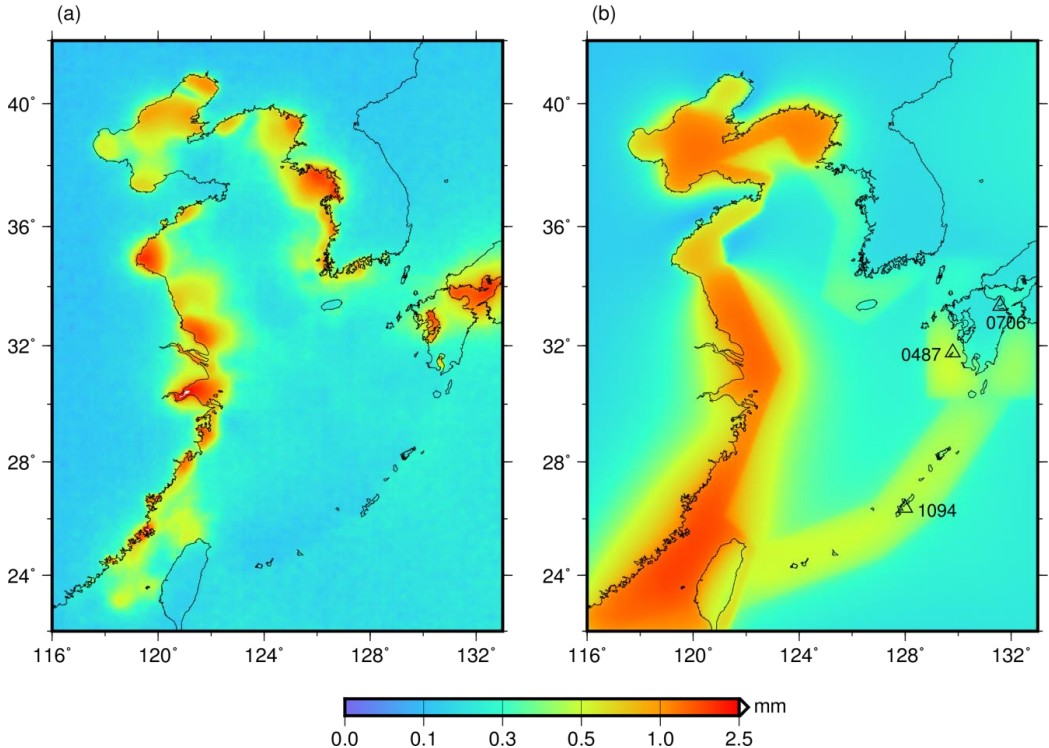

**Figure 3 (a)** The standard deviation of $M_2$ vertical OTL displacements, computed using the nine ocean tide models and an elastic PREM Green's function. **(b)** The $M_2$ vertical OTL errors per grid point according to equation (4), using the RMS errors in NAO99Jb based on comparisons with tide gauges, and an elastic PREM Green's function.

Using equation 4 and inputting the NAO99Jb RMS errors per sub-area, the $M_2$ vertical OTL displacement errors at each point of a 1/8° grid were computed and are shown in Figure 3b. It can be seen that the largest errors of 1-2 mm are for the points falling within the eastern China sub-area, but these can be explained by the NAO99Jb model having a fairly large RMS error of 11.7 cm for this sub-area, and this has the largest influence on the OTL displacement there. For the rest of the ECS region, notably where most of the GPS sites are located, the OTL errors arising from NAO99Jb model RMS errors are no more than ~0.5 mm, even for sites on the east of Kysushu where the inter-model OTL STDs are large (~2.5 mm).

To provide a more detailed indication of the influence on the OTL of the NAO99Jb ocean tide model errors from each of the defined sub-areas, three GPS sites (0487, 0706 and 1094) are considered, located on the east and west of Kyushu and on the Ryukyu Islands (Figure 3b). The contribution of





each sub-area to both the OTL displacement and its accompanying error are shown in Table 3, which provides further clarification that the local ocean tides are the principal contributor to the OTL displacements, as well as the OTL errors. The large effect from the 'other water areas' is mainly due to their vast area, although most of this is far from our study area and will have no impact on regional comparison of Earth models. The Kanmen Straits and eastern China, where NAO99Jb performs relatively poorly, have little effect on the OTL displacements at these sites, with contributions to the OTL amplitude and error of only 1.0-1.5 mm and less than 0.1 mm, respectively. Furthermore, the effect of the ocean tide model errors from these two sub-areas is no more than 0.13 mm for all three sites. These computations were repeated for all the GPS sites, and only three of the 102 GPS sites had a total OTL prediction error greater than 0.5 mm. It can therefore be concluded that the OTL displacements computed using the NAO99Jb ocean model are suitable for investigating possible anelasticity effects in the ECS region.

**Table 3** The contribution of the defined water sub-areas in Figure 2 to the $M_2$ vertical OTL displacement amplitudes and the resulting errors at GPS sites 0487, 0706 and 1094 according to equation 4 and using the NAO99Jb model and its RMS errors.

| Area | $M_2$ vertical OTL amp (mm) | | | $M_2$ vertical OTL error (mm) | | |
|---|---|---|---|---|---|---|
| | 0487 | 0706 | 1094 | 0487 | 0706 | 1094 |
| Eastern China | 1.18 | 0.99 | 1.48 | 0.11 | 0.09 | 0.13 |
| Ryukyu Islands | 1.67 | 0.92 | 9.46 | 0.07 | 0.04 | 0.42 |
| KyushuW | 8.04 | 1.11 | 0.34 | 0.41 | 0.06 | 0.02 |
| Ariake Sea | 0.42 | 0.34 | 0.03 | 0.01 | 0.01 | 0.00 |
| KyushuE | 1.10 | 1.36 | 0.21 | 0.06 | 0.08 | 0.01 |
| KyushuN | 0.53 | 0.81 | 0.08 | 0.02 | 0.04 | 0.00 |
| Seto Inland Sea | 0.34 | 3.28 | 0.05 | 0.01 | 0.14 | 0.00 |
| Kanmen Straits | 0.00 | 0.01 | 0.00 | 0.00 | 0.00 | 0.00 |
| Korea | 0.18 | 0.15 | 0.13 | 0.02 | 0.01 | 0.00 |
| Other Water Areas | 7.78 | 5.81 | 13.47 | 0.13 | 0.17 | 0.13 |
| Total | 18.33 | 10.54 | 22.53 | 0.46 | 0.25 | 0.46 |

## 4 Kinematic GPS estimation of OTL displacement

Using the NASA GNSS-Inferred Positioning System (GIPSY) software in kinematic precise point positioning (PPP) mode, Penna et al. (2015) showed for sites in western Europe with at least 2.5 years of GPS data (4 years recommended), that vertical OTL displacements may be estimated with a precision of about 0.2-0.4 mm. We apply the same approach for GPS sites in the ECS region. In order to assess the accuracy and precision of the OTL displacements, particularly to check that the tuned coordinate and tropospheric delay process noise values for western Europe are applicable for the ECS region, we insert an artificial harmonic displacement per GPS site. We then assess how well it is recovered from the kinematic PPP GPS processing, as per Penna et al. (2015) but in the time series used for the final OTL displacement estimation rather than as a preliminary investigation step.

### 4.1 GPS data source

All available continuous GPS data in the ECS region were collated for the window 2013.0-2017.0,



with the distribution of the 102 sites used shown in Figure 1. These comprised 96 sites from the GPS Earth Observation Network (GEONET), which all had at least 95% data availability throughout the 4-year window considered, and are located mainly on the Ryukyu Islands and Kyushu. We also collated data from six International GNSS Service (IGS) sites in China and Korea, although two sites

(SHAO and YONS) only had 2.5 years of data. On the Ryukyu Islands and along the coast of Kyushu, the sites exhibit detectability ratios of greater than one, with the median value being 2.1, although close to the Seto Island Sea the ratio reduces to less than one. The data spans of at least 2.5 and typically 4 years are sufficient to separate the different major tidal constituents robustly according to the Rayleigh criterion.

### 4.2 Data analysis strategy

Full details of the GPS data processing strategy used are provided in Penna et al. (2015): in summary it is as follows. Daily, 30-hour, kinematic PPP GPS solutions were generated for each site using GIPSY version 6.4 software with Jet Propulsion Laboratory (JPL) reprocessed version 2.1 fiducial

satellite orbits, Earth orientation parameters and 30 s satellite clocks held fixed in the IGb08 reference frame. A priori hydrostatic and wet zenith tropospheric delays from the European Centre for Medium-Range Weather Forecasts reanalysis product were used, with residual zenith tropospheric delays estimated every 5 min (applying a process noise of 0.1 mm/$\sqrt{s}$), together with north-south and east-west tropospheric gradients. The VMF1 gridded mapping function was used with an elevation cut-

off of 10°, and corrections were applied for solid Earth and pole tides according to the IERS Conventions 2010 (Petit and Luzum, 2010), along with IGS satellite and receiver antenna phase centre variation corrections. Ambiguities were fixed to integers according to the approach of Bertiger et al. (2010). Receiver coordinates were estimated every 5 min, with a coordinate process noise of 3.2 mm/$\sqrt{s}$ applied. OTL displacement was modelled using the IERS Conventions (2010) hardisp

routine, based on amplitudes and phase lags generated using the NLOADF software with the NAO99Jb model (augmented in the rest of the world with the FES2014b model) and a PREM elastic Green's function, computed in the centre of mass of the solid Earth and oceans (CM) frame to be compatible with the JPL orbits. In each daily solution, an artificial 13.96 hour harmonic signal of 3.0 mm amplitude was introduced in each of the east, north and vertical components, with the phase

referenced to zero defined at GPS time frame epoch J2000.

The estimated coordinates at 5 min resolution within the central 24 hours of the daily 30-hour kinematic PPP GPS solutions (which ran from 21:00 the previous day to 03:00 the next day) were averaged in non-overlapping, 30 min bins, then concatenated to form coordinate time series.

Harmonic analysis was then undertaken using UTide to estimate the residual $M_2$ vertical OTL displacement signal per site, and also a 13.96 hour harmonic was estimated to assess how well the introduced 3.0 mm amplitude artificial signal could be recovered. The resulting UTide formal errors were 0.1-0.2 mm.

### 4.3 Results

The $M_2$ vertical OTL residual phasors extracted from the harmonic analysis are shown in Figure 4, as well as the artificial 13.96 hour harmonic signal residual phasors. It can be seen from Figure 4 that on the Ryukyu Islands and in the west coastal area of Kyushu the $M_2$ vertical OTL GPS-observed minus model discrepancies (residuals) can reach over 1.5 mm, corresponding to about 7% of the total

loading signal. The typical magnitudes of phasor differences between the recovered and original





artificial 13.96 hour harmonic signals are 0.2-0.3 mm, providing an indication of the accuracy level of our GPS-observed $M_2$ vertical OTL displacements, and indicating that the optimal process noise values found for western Europe by Penna et al. (2015) are also applicable to the ECS region. Since the ocean tide error of NAO99Jb maps to only an error of 0.2-0.5 mm for the predicted $M_2$ vertical OTL displacement values across the Ryukyu Islands and Kyushu (Figure 3b), it can be concluded that the 1.5 mm discrepancies must be dominated by errors in the elastic PREM Green's function.

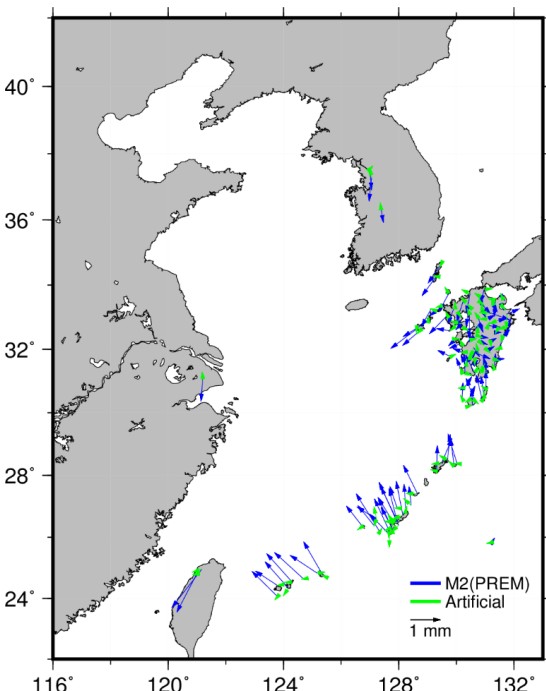

**Figure 4** Phasor differences (in blue) between the GPS-observed $M_2$ vertical OTL displacements and the predictions computed using the NAO99Jb regional ocean tide model (augmented elsewhere globally with FES2014b) and an elastic PREM Green's function. Also shown (in green) are the phasor differences between the recovered and original artificial ~13.96 hour harmonic vertical displacement signal of 3.0 mm amplitude.

## 5 Optimal Green's function for the East China Sea region

As Green's functions essentially depend on the material properties of the adopted Earth models, the improvement of the agreement between GPS-observed and predicted OTL values could be expected by modifying these properties, and those of the asthenosphere have been demonstrated to be important (Bos et al. 2015). In order to compute an optimal Green's function for the ECS region, instead of PREM we consider the more recent S362ANI Earth model (Kustowski et al. 2008), which is a transversely isotropic seismic tomographic model for the upper mantle. For the computation, the mean shear velocity of S362ANI was prepared using an area centred on the oceanic region of interest, between longitudes 122° and 133° east and latitudes 23° and 35° north. For the density and compressional velocity, S362ANI only provides global mean profiles. In our work, the asthenosphere is defined a priori to be between depths of 80 and 220 km with a $Q$ of 70. Following a similar method





to Bos et al. (2015), we vary the depths of the top (D1) and bottom (D2) of the asthenosphere of S362ANI, and the amount of anelastic dispersion ($Q$) in this layer. For each combination of these three parameters, a new Green's function was computed via the load Love number formulation. While computing the load Love numbers, we transformed the shear modulus from the reference period (1 s) to the period of harmonic $M_2$ using the relation formula given by Dahlen and Tromp (1998), with the $Q$ value assumed constant over this range of periods. The $Q$ value in the other layers is at least twice that of the asthenosphere so the frequency dependence will be smaller, but to be consistent the elastic properties were also transformed to the period of harmonic $M_2$. However, these $Q$ values were not varied in our inversion. New Green's functions were then derived and used to predict the $M_2$ vertical OTL values using the NAO99Jb ocean tide model. This transformation produces complex-valued shear moduli and therefore complex-valued Green's functions but the imaginary part is less than 5% of the real part, see Bos et al. (2015), and can be neglected. The optimal Green's function is considered to be that which minimises the sum of the squared misfits between the observed and predicted OTL phasor values using all the GPS sites.

The optimal Green's function has been obtained when $Q$ is 90 (corresponding to a reduction of the shear modulus of about 7.6% at the $M_2$ period), and the estimated values of D1 and D2 are 40 and 220 km, respectively, implying an asthenosphere extending to shallower depths than its original definition for this region in S362ANI. To indicate that the optimal Green's function refers to the period of $M_2$, the label "M2" is added to the S362ANI name, along with the prefix 'mod' to denote that it has been modified. Figure 5 compares the phasor differences of the predicted $M_2$ vertical OTL displacements computed using NAO99Jb and the Green's functions PREM and mod_S362ANI_M2, with respect to the GPS-observed values. It can be seen that for mod_S362ANI_M2 the discrepancies have been substantially reduced on the Ryukyu Islands, where the ~1.5 mm discrepancies typically reduce to less than 0.8 mm, and the RMS agreement has been improved by about 0.3 mm (~0.7 mm reducing to ~0.4 mm) compared to the elastic PREM Green's function (Table 4). Therefore, the large influence of the asthenosphere has been validated.





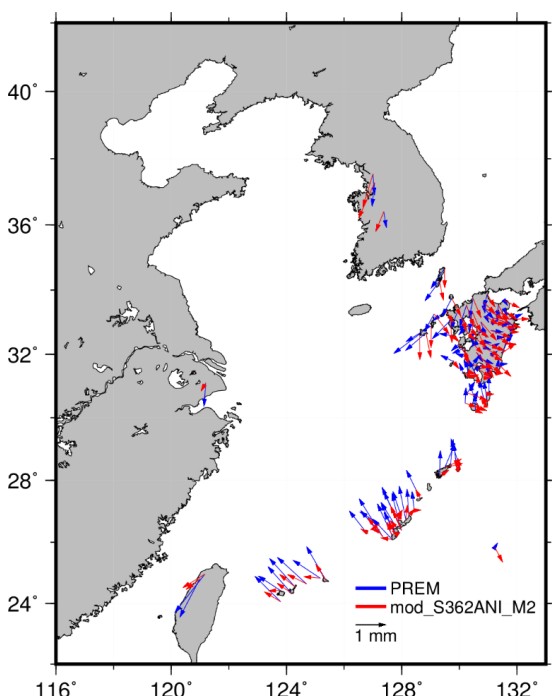

**Figure 5** Phasor differences between our GPS-observed $M_2$ vertical OTL displacements and the predictions computed using the NAO99Jb regional ocean tide model (augmented elsewhere globally with FES2014b) and the elastic PREM (blue phasors) and mod_S362ANI_M2 (red phasors) Green's functions.

For completeness, we also compare the GPS-observed $M_2$ vertical OTL displacements with the predictions computed using the Green's function based on the a priori definition of S362ANI for the ECS region, as well as the ones based on PREM and S362ANI with their anelastic dispersion effect directly corrected, termed PREM_M2 and S362ANI_M2, respectively. The RMS, minimum and

10   maximum values of the $M_2$ vertical OTL phasor differences per Green's function are shown in Table 4. It can be seen that using the elastic S362ANI Green's function reduces the overall RMS by about 0.1 mm compared to the elastic PREM Green's function, which could be explained by applying the regional mean shear velocity. The RMS agreement can be further improved by correcting the anelastic dispersion effect (PREM_M2 and S362ANI_M2), and their results are quite similar to the optimal

15   mod_S362ANI_M2 Green's function, which provides a simple way to improve predicted OTL displacements instead of performing the complex numerical optimisation scheme each time.





**Table 4** Statistics (in mm) of the phasor differences between the GPS-observed and predicted $M_2$ vertical OTL displacements using the NAO99Jb regional ocean tide model (augmented elsewhere globally with FES2014b) and various Green's functions.

| Green's function | The whole ECS region | | | Ryukyu Islands | | |
|---|---|---|---|---|---|---|
| | Minimum | Maximum | RMS | Minimum | Maximum | RMS |
| PREM | 0.08 | 1.59 | 0.53 | 0.63 | 1.54 | 0.74 |
| S362ANI | 0.06 | 1.45 | 0.45 | 0.50 | 1.45 | 0.66 |
| PREM_M2 | 0.10 | 1.12 | 0.41 | 0.25 | 1.12 | 0.47 |
| S362ANI_M2 | 0.01 | 1.39 | 0.40 | 0.13 | 0.95 | 0.37 |
| mod_S362ANI_M2 | 0.09 | 1.26 | 0.39 | 0.17 | 1.02 | 0.41 |

## 6 Conclusions

By introducing the detectability ratio for the asthenospheric anelasticity effects and considering the distribution of the available GPS sites, the ECS region was selected as a potential area to observe the anelastic dispersion in the asthenosphere. Using an inter-comparison of eight recent global (DTU10,
EOT11a, FES2014b, GOT4.10c, HAMTIDE11a, NAO99b, OSU12, TPXO9-Atlas) and one regional (NAO99Jb) models and a validation with tide gauges, NAO99Jb has been demonstrated to be the most accurate tide model in the region. In the open sea areas NAO99Jb is slightly worse than the other ocean tide models, due to the assimilation of more satellite altimetry data in the latter, but this does not outweigh the benefits of forcing the NAO99Jb model to fit a large amount of tide gauge
observations. We quantified the impact of the errors in NAO99Jb on the predicted OTL values, based on the RMS difference between NAO99Jb and the tide gauge observations. Compared to the approach of using the STD of predicted OTL displacements as the error contribution of the ocean tide models, this method can allow for systematic biases shared by the models, so the outputs are more realistic. For the GPS sites located in Japan, the errors in NAO99Jb result in $M_2$ vertical OTL displacement
errors of 0.2-0.5 mm.

We then estimated the $M_2$ vertical OTL displacements for 102 sites around the ECS using GPS with typical accuracy of 0.2-0.3 mm. On the Ryukyu Islands and in the west coastal area of Kyushu, the discrepancies between GPS-observed and predicted values can reach over 1.5 mm when using the
NAO99Jb tide model and the purely elastic PREM Green's function. However, the discrepancies cannot be explained by the sum of the remaining errors due to ocean tide models and the uncertainty in the GPS observations themselves. Given that the observations are sensitive to the elastic properties of the asthenosphere, we estimated an optimal Green's function by varying the depth and thickness of the asthenosphere of the S362ANI Earth model and its $Q$ values, which were used to model the
anelastic dispersion effect during the computations. A reduction of about 7.6% of the shear modulus has been confirmed to produce the best agreement, which reduces the discrepancies to no more than 0.8 mm on the Ryukyu Islands, clearly demonstrating the importance of considering the anelastic properties of the asthenosphere.

This paper has confirmed the importance of considering the asthenospheric anelasticity effects observed by Bos et al. (2015). It is necessary to incorporate the dissipation effects for the Green's functions based on seismic Earth models: use of elastic parameters at 1 s period is insufficient. The





PREM_M2 Green's function is near-optimal for the ECS region and western Europe, and represents a sensible compromise with global applicability so is therefore a pragmatic choice for OTL prediction in geodetic analysis. For sites in areas where the detectability ratio exceeds one shown in Figure 1a, or where the highest accuracy is demanded, a regional anelastic Green's function calculated directly 5    from a laterally-varying Earth model such as S362ANI should be considered.

*Data availability*. GPS data were obtained from the International GNSS Service (www.igs.org) and by request from the GPS Earth Observation Network (GEONET) of the Geospatial Information 10   Authority of Japan (GSI) (http://datahouse1.gsi.go.jp/terras/terras_english.html). The ocean tide models used were those provided in the SPOTL software version 3.3.0.2 distribution (https://igppweb.ucsd.edu/~agnew/Spotl/spotlmain.html), except FES2014b was obtained from https://www.aviso.altimetry.fr/en/data/products/auxiliary-products/global-tide-fes.html, OSU12 from https://earthsciences.osu.edu/divisions/geodeticoceantides/OSU12v1.0/, TPXO9-Atlas from 15   http://volkov.oce.orst.edu/tides/tpxo9_atlas.html, and GOT4.10c was provided by its author Richard Ray, NASA-Goddard Space Flight Center via personal communication. The Earth models were obtained from http://ds.iris.edu/ds/products/emc-referencemodels/, JPL orbits and clocks from ftp://sideshow.jpl.nasa.gov, tide gauge data from the Japan Oceanographic Data Centre (https://www.jodc.go.jp/jodcweb/) and the University of Hawaii Sea Level Center 20   (https://uhslc.soest.hawaii.edu/).

## Appendix A: Mass loading Green's function based on PREM

Here we list the Green's functions for the vertical (radial) and horizontal displacement based on the 25   isotropic version of the Preliminary Reference Earth Model (PREM) used in this study. This is still one of the most-used Earth models for loading computations. The elastic properties have been derived from seismic observations and are valid at the reference period of 1 s. More details of the computations of the load Love numbers are presented by Bos and Scherneck (2013). To include the anelastic dispersion effect, the values of the shear modulus were converted to the period of $M_2$ with 30   a constant absorption band assumed as described by Bos et al. (2015). The bulk modulus has a much higher quality factor $Q$ and is assumed not to be affected. After modifying the shear modulus, the load Love numbers were computed in the same manner. The first column contains the angular distance between the point load and the station under interest. The second and third columns contain the scaled Green's functions for the vertical and horizontal displacement respectively for the purely elastic 35   PREM. Columns four and five are the same as the previous two but for the one with the anelastic dispersion effect corrected (PREM_M2). Due to the smallness of the imaginary part of the Green's function of PREM_M2, we only list the real part. All the Green's functions are computed in the CM frame which has its origin at the centre of mass of the solid Earth plus ocean tides. To obtain Green's functions in the CE frame (origin at the centre of mass of the solid Earth alone), the quantity 40   $a \cos\theta / m_E$ should be added to the radial Green's function $u_r$ and $a \sin\theta / m_E$ subtracted from the horizontal Green's function $u_\theta$, where $a$ and $m_E$ are respectively the mean radius and mass of the Earth.





**Table A1** Green's function per kilogram of mass load for the radial and horizontal displacements based on PREM with units of m/kg

| θ (deg) | PREM | | PREM_M2 | |
|---|---|---|---|---|
| | $u_r \times 10^{12}(a\theta)$ | $u_\theta \times 10^{12}(a\theta)$ | $u_r \times 10^{12}(a\theta)$ | $u_\theta \times 10^{12}(a\theta)$ |
| 0.0001 | -42.174 | -12.844 | -42.532 | -12.866 |
| 0.0010 | -41.982 | -12.844 | -42.340 | -12.866 |
| 0.0100 | -40.081 | -12.808 | -40.432 | -12.829 |
| 0.0200 | -37.998 | -12.699 | -38.343 | -12.720 |
| 0.0300 | -35.965 | -12.521 | -36.303 | -12.540 |
| 0.0400 | -33.999 | -12.280 | -34.332 | -12.297 |
| 0.0600 | -30.335 | -11.640 | -30.661 | -11.652 |
| 0.0800 | -27.111 | -10.853 | -27.434 | -10.860 |
| 0.1000 | -24.375 | -10.000 | -24.700 | -10.001 |
| 0.1600 | -18.950 | -7.662 | -19.310 | -7.655 |
| 0.2000 | -17.100 | -6.596 | -17.499 | -6.593 |
| 0.2500 | -15.919 | -5.832 | -16.373 | -5.843 |
| 0.3000 | -15.379 | -5.511 | -15.889 | -5.543 |
| 0.4000 | -14.949 | -5.483 | -15.556 | -5.566 |
| 0.5000 | -14.676 | -5.660 | -15.351 | -5.797 |
| 0.6000 | -14.375 | -5.798 | -15.091 | -5.981 |
| 0.8000 | -13.691 | -5.854 | -14.420 | -6.102 |
| 1.0000 | -12.984 | -5.729 | -13.660 | -5.997 |
| 1.2000 | -12.298 | -5.527 | -12.890 | -5.780 |
| 1.6000 | -11.045 | -5.065 | -11.455 | -5.229 |
| 2.0000 | -9.967 | -4.609 | -10.229 | -4.671 |
| 2.5000 | -8.851 | -4.088 | -8.989 | -4.050 |
| 3.0000 | -7.959 | -3.633 | -8.025 | -3.531 |
| 4.0000 | -6.689 | -2.905 | -6.689 | -2.748 |
| 5.0000 | -5.895 | -2.375 | -5.874 | -2.210 |
| 6.0000 | -5.407 | -1.991 | -5.379 | -1.832 |
| 7.0000 | -5.115 | -1.715 | -5.084 | -1.564 |
| 8.0000 | -4.946 | -1.515 | -4.916 | -1.375 |
| 9.0000 | -4.854 | -1.369 | -4.826 | -1.239 |
| 10.0000 | -4.809 | -1.258 | -4.784 | -1.138 |
| 12.0000 | -4.791 | -1.094 | -4.772 | -0.992 |
| 16.0000 | -4.833 | -0.820 | -4.823 | -0.749 |
| 20.0000 | -4.851 | -0.497 | -4.844 | -0.448 |
| 25.0000 | -4.779 | 0.015 | -4.771 | 0.049 |
| 30.0000 | -4.587 | 0.645 | -4.577 | 0.672 |
| 40.0000 | -3.890 | 2.214 | -3.875 | 2.235 |
| 50.0000 | -2.928 | 4.049 | -2.910 | 4.069 |
| 60.0000 | -1.860 | 5.931 | -1.840 | 5.951 |
| 70.0000 | -0.753 | 7.661 | -0.734 | 7.681 |
| 80.0000 | 0.395 | 9.103 | 0.411 | 9.120 |
| 90.0000 | 1.617 | 10.174 | 1.629 | 10.187 |





| | | | |
|---|---|---|---|
| 100.0000 | 2.949 | 10.831 | 2.955 | 10.837 |
| 110.0000 | 4.405 | 11.049 | 4.405 | 11.049 |
| 120.0000 | 5.977 | 10.816 | 5.970 | 10.811 |
| 130.0000 | 7.630 | 10.125 | 7.615 | 10.117 |
| 140.0000 | 9.306 | 8.974 | 9.284 | 8.964 |
| 150.0000 | 10.933 | 7.367 | 10.904 | 7.357 |
| 160.0000 | 12.428 | 5.318 | 12.394 | 5.310 |
| 170.0000 | 13.709 | 2.851 | 13.669 | 2.846 |
| 180.0000 | 14.696 | 0.000 | 14.653 | 0.000 |

## Appendix B: Mass loading S362ANI Green's functions for the East China Sea

Here we list the Green's functions of the vertical (radial) and horizontal displacement based on the S362ANI Earth model of Kustowski et al. (2008). This model provides horizontal and vertical shear velocities (transversely isotropic) on a regular longitude/latitude grid for various depths. For each depth layer between longitudes 122° and 133° east and latitudes 23° and 35° north, we computed averaged shear velocities. Together with the density and compressional velocities of the STW105 model (Kustowski et al., 2008), this provides us enough information to compute load Love numbers. These load Love numbers were computed in the same manner as those for PREM in Appendix A. Columns 2 and 3 contain the Green's functions based on the elastic properties valid at 1 s. Columns 4 and 5 are based on the elastic properties converted to the period of harmonic M₂. For the Green's functions listed in columns 6 and 7 the depths of the top and bottom of the asthenosphere have also been changed to 40 and 220 km respectively while the quality factor $Q$ has been increased to 90. Due to the smallness of the imaginary part of the Green's function, we only list the real part. Again, the Green's functions are given for the CM frame.

**Table B1** Green's function per kilogram of mass load for the radial and horizontal displacements based on S362ANI, S362ANI_M2 and mod_S362ANI_M2 with units of m/kg

| $\theta$ (deg) | S362ANI | | S362ANI_M2 | | mod_S362ANI_M2 | |
|---|---|---|---|---|---|---|
| | $u_r \times 10^{12}(a\theta)$ | $u_\theta \times 10^{12}(a\theta)$ | $u_r \times 10^{12}(a\theta)$ | $u_\theta \times 10^{12}(a\theta)$ | $u_r \times 10^{12}(a\theta)$ | $u_\theta \times 10^{12}(a\theta)$ |
| 0.0001 | -42.198 | -12.851 | -42.923 | -12.894 | -42.933 | -12.897 |
| 0.0010 | -42.013 | -12.851 | -42.736 | -12.894 | -42.748 | -12.897 |
| 0.0100 | -40.183 | -12.818 | -40.888 | -12.860 | -40.913 | -12.863 |
| 0.0200 | -38.182 | -12.714 | -38.868 | -12.755 | -38.907 | -12.759 |
| 0.0300 | -36.228 | -12.543 | -36.898 | -12.582 | -36.950 | -12.588 |
| 0.0400 | -34.341 | -12.312 | -34.995 | -12.348 | -35.060 | -12.355 |
| 0.0600 | -30.831 | -11.699 | -31.458 | -11.726 | -31.548 | -11.739 |
| 0.0800 | -27.748 | -10.947 | -28.356 | -10.965 | -28.469 | -10.986 |
| 0.1000 | -25.141 | -10.134 | -25.739 | -10.144 | -25.871 | -10.173 |
| 0.1600 | -20.004 | -7.934 | -20.615 | -7.932 | -20.782 | -7.990 |
| 0.2000 | -18.263 | -6.955 | -18.906 | -6.959 | -19.074 | -7.036 |
| 0.2500 | -17.141 | -6.278 | -17.830 | -6.302 | -17.977 | -6.395 |
| 0.3000 | -16.596 | -6.014 | -17.329 | -6.065 | -17.435 | -6.162 |
| 0.4000 | -16.053 | -6.016 | -16.852 | -6.122 | -16.846 | -6.196 |
| 0.5000 | -15.620 | -6.155 | -16.455 | -6.311 | -16.330 | -6.327 |



| 0.6000 | -15.158 | -6.225 | -16.002 | -6.420 | -15.780 | -6.364 |
|---|---|---|---|---|---|---|
| 0.8000 | -14.190 | -6.126 | -14.993 | -6.367 | -14.665 | -6.178 |
| 1.0000 | -13.259 | -5.855 | -13.967 | -6.099 | -13.630 | -5.834 |
| 1.2000 | -12.406 | -5.529 | -12.999 | -5.746 | -12.706 | -5.460 |
| 1.6000 | -10.952 | -4.892 | -11.319 | -5.002 | -11.157 | -4.773 |
| 2.0000 | -9.787 | -4.343 | -9.984 | -4.337 | -9.928 | -4.202 |
| 2.5000 | -8.643 | -3.777 | -8.708 | -3.663 | -8.727 | -3.621 |
| 3.0000 | -7.761 | -3.318 | -7.759 | -3.140 | -7.807 | -3.152 |
| 4.0000 | -6.547 | -2.630 | -6.502 | -2.411 | -6.551 | -2.460 |
| 5.0000 | -5.814 | -2.158 | -5.768 | -1.949 | -5.800 | -1.993 |
| 6.0000 | -5.375 | -1.833 | -5.335 | -1.646 | -5.354 | -1.678 |
| 7.0000 | -5.117 | -1.609 | -5.082 | -1.443 | -5.093 | -1.465 |
| 8.0000 | -4.970 | -1.452 | -4.939 | -1.304 | -4.946 | -1.319 |
| 9.0000 | -4.889 | -1.338 | -4.862 | -1.207 | -4.866 | -1.216 |
| 10.0000 | -4.848 | -1.251 | -4.825 | -1.133 | -4.828 | -1.140 |
| 12.0000 | -4.827 | -1.114 | -4.810 | -1.019 | -4.812 | -1.022 |
| 16.0000 | -4.854 | -0.852 | -4.845 | -0.789 | -4.845 | -0.790 |
| 20.0000 | -4.861 | -0.523 | -4.854 | -0.481 | -4.854 | -0.481 |
| 25.0000 | -4.782 | -0.002 | -4.775 | 0.027 | -4.775 | 0.027 |
| 30.0000 | -4.588 | 0.634 | -4.579 | 0.656 | -4.579 | 0.656 |
| 40.0000 | -3.891 | 2.206 | -3.879 | 2.224 | -3.878 | 2.223 |
| 50.0000 | -2.931 | 4.041 | -2.916 | 4.059 | -2.915 | 4.058 |
| 60.0000 | -1.863 | 5.922 | -1.847 | 5.942 | -1.846 | 5.939 |
| 70.0000 | -0.756 | 7.653 | -0.741 | 7.672 | -0.740 | 7.669 |
| 80.0000 | 0.392 | 9.095 | 0.405 | 9.112 | 0.406 | 9.108 |
| 90.0000 | 1.614 | 10.167 | 1.624 | 10.180 | 1.624 | 10.176 |
| 100.0000 | 2.945 | 10.824 | 2.950 | 10.831 | 2.950 | 10.827 |
| 110.0000 | 4.401 | 11.043 | 4.401 | 11.045 | 4.400 | 11.041 |
| 120.0000 | 5.973 | 10.811 | 5.967 | 10.808 | 5.965 | 10.804 |
| 130.0000 | 7.625 | 10.121 | 7.613 | 10.114 | 7.610 | 10.110 |
| 140.0000 | 9.301 | 8.970 | 9.283 | 8.962 | 9.279 | 8.959 |
| 150.0000 | 10.927 | 7.365 | 10.903 | 7.356 | 10.899 | 7.353 |
| 160.0000 | 12.422 | 5.316 | 12.393 | 5.310 | 12.388 | 5.308 |
| 170.0000 | 13.702 | 2.850 | 13.669 | 2.846 | 13.663 | 2.845 |
| 180.0000 | 14.688 | 0.000 | 14.653 | 0.000 | 14.646 | 0.000 |

*Author contributions*. PJC, NTP and JW devised the study, NTP undertook GIPSY processing of the GPS data, JW carried out analysis of the tide gauge observations, ocean tide loading displacements and GPS coordinate time series under the supervision of NTP and PJC, and MSB computed the elastic and anelastic Green's functions. All authors contributed to the discussion of the results and writing of the manuscript.

*Competing interests*. The authors declare that they have no conflicts of interest.

*Special issue statement*. This article is part of the special issue "Developments in the science and



history of tides (OS/ACP/HGSS/NPG/SE inter-journal SI)". It is not associated with a conference.

*Acknowledgements.* All data providers listed in the Data Availability statement are thanked, as well as Duncan Agnew, Daniel Codiga and NASA JPL for providing the SPOTL, UTide and GIPSY software packages, respectively. The figures were generated using the Generic Mapping Tools software (Wessel et al., 2013). This work was funded by the award of a Chinese Scholarship Council 201606710069 to JW and UK Natural Environment Research Council grant NE/R010234/1 to PJC and NTP, in partnership with MSB.

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
