# Peer review of "Asthenospheric anelasticity effects on ocean tide loading in the East"

_Solid Earth, 2019_

## Referee Comment (RC1) · Richard Ray (Referee) · 6 Oct 2019

This paper is a follow-up to earlier work by this group (e.g., Bos et al., 2015) on anelastic effects in tidal loading, a long-sought goal of the earth-tide community. It is important work, and this is a carefully done study. Figure 1 will be immediately useful to many researchers. I have only minor comments, with the exception of my first point, which might be more important.

(1) I want to bring up a possible systematic error which, if not addressed here, needs to be addressed at some point by this group as they continue to do these kinds of studies. I think it could cause errors of a few percent, which could be significant. Specifically,

I'm beginning to think these very precise geodetic applications need to account for the variable density of seawater – that rho in Eqns (1-2) should stay inside the integral sign.

My 2013 paper (already cited by the authors) has a section (Section 2.4) on this in the context of bottom pressure measurements, and elsewhere (Ray et al., JGR, 2009 – doi:10.1029/2009JC005362) we've seen evidence that GRACE may be sensitive to it (although at this stage, errors in the ocean tide models themselves still dominate GRACE sensitivity). A version of Duncan Agnew's SPOTL package does account for variable seawater density, but initially it was using the density at the ocean surface (I'm not sure about his most recent version). But a rising tide is caused by convergence through the whole water column, so for this reason, in the 2009 paper, I used the mean column density. After discussions with Chris Garrett, I think compressibility is also involved, and my 2013 paper uses an expression that Chris worked out (involving the speed of sound). In practice, for the ocean we currently have, Chris's expression is numerically about equal to the density at the seafloor.

I won't insist the authors investigate this effect here in this paper, but before they publish too many of these kinds of studies, it would be worth looking into.

Other minor things:

(2) Abstract, line 13, recommend inserting "the regional model" before NAO99Jb, because many readers, even in the tide community, may see NAO99 and think it refers to Matsumoto's global altimeter-based model.

(3) Next line: "the most accurate". Well, this is risky because who knows if someone has developed another regional model here. I'd say "an accurate" – but it's up to the authors.

(4) In Abstract, and also page 15, bottom, authors quote 1.5 mm and 0.8 mm. I don't understand this. From Table 4, this looks to be "comparing apples and oranges". One is a maximum error and the other is RMS. Seems misleading, unless I'm just not following

where they get these numbers.

(5) Page 13, line 5: Dahlen & Tromp is a massive book. I and many readers would appreciate your quoting the page number or even the Eqn number you're using when you cite this book.

(6) Page 2, line 36, where GPS is assigned an error of 0.3 mm. I don't accept this, because surely the errors in GPS are dependent on the length of the time series.

Richard Ray

---

## Referee Comment (RC2) · Anonymous Referee #2 · 7 Oct 2019

Review of "Asthenospheric anelasticity effects on ocean tide loading in the East China Sea region observed with GPS" by Wang et al.

General comments

This paper covers work on extending the methods and results obtained by the authors in their 2 papers in 2015 (Penna et al. and Bos et al.). These previous papers showed the importance of including the effects of anelasticity in the OTL model computations for continuous GPS sites in western Europe. The present paper demonstrates that anelasticity is also important in the East China Sea region. This is a completely different type of tectonic area. This is therefore a very important paper and shows that

anelasticity in the asthenosphere should be considered in OTL computations at most sites, within a few hundred kilometres of the coast, around the world. The paper again shows that the continuing dense network of GPS stations around the world provides a very useful data resource for tidal research, as well as the more common use for studying long term crustal movements. The methods used and the results are very clearly described in the paper and I strongly recommend that this paper should be published in this journal. A few mostly relatively minor comments are given below.

Specific comments (1) The first part of the introduction gives a useful summary of previous work on using CGPS for tidal research. On page 2, line 9 it would be useful to mention that the GPS results from Yuan et al. (2013) were used by Lau, Mitrovica, . . .(Nature September 2017) to look for lateral variations in body tide models of the lower mantle. (2) Page 3, line 16 (also page 7, line 5). It would be worth pointing out that Baker and Bos (2003, Fig. 9) used tidal gravity observations in Wuhan, China, to show that there are major problems with the earlier set of FES ocean tide models in that area. (3) Page 5, line 32. STDs of the phasor differences. . ..should be STDs of the amplitudes of the phasor differences. (4) In Figures 4 and 5, it is very difficult to see all the phasors in Kyushu and the maps look very jumbled in that area. It is not easy to get round this problem. Maybe it would be better to reduce the number of sites/phasors and say that for clarity only 50% (or whatever) of the phasors are shown. (5) In Figure 5, the final (red) phasors still show some correlations along the Ryukyu islands. This implies that there is still some information left in these residuals. The authors may want to comment on this. (6) Page 15, line 32. It is stated that the discrepancies are less than 0.8 mm on the Ryukyu Islands. This is not consistent with Table 4 on the same page.

---

## Referee Comment (RC3) · Duncan Agnew (Referee) · 8 Oct 2019

Review of SE MS "Asthenospheric anaelasticity effects on ocean tide loading in the East China Sea..." by Wang *et al.*

This paper looks at observed $M_2$ tides in the vertical displacement of continuous GPS stations and compares them with model tides, especially model tides for ocean loading. The paper starts with an excellent depiction of where the differences in loading from anelastic and elastic Green functions are largest; the authors choose to look at the region around the East China Sea.

The authors examine a wide range of ocean tide models for this region, including many global models, though in the end they focus on predicted load tides from the regional model of Matsumoto *et al.* (2000), NAO99Jb, which they choose based on their own tidal analysis of sea level from 75 tide gauges. (They also use a global model, but this will make only a small contribution). They find that NAO99Jb provides a much better RMS fit (Table 2) to the observed $M_2$ ocean tide at these gauges than any of the global models. Computing the ocean loads with NAO99Jb and a PREM Green function, they find misfits to the observed $M_2$ vertical displacement tide. These can be reduced by using a local crust-mantle model instead of PREM, and also by allowing for anelastic dispersion. Their final choice is a modification of the local model with a shallower asthenosphere.

The extraction of observed $M_2$ tides from the data, the computation of Green functions from the different models, and the load computation have been done well: an impressive amount of work. Despite this, I think the paper requires major revision, because some of the choices made are not well justified, and also because the paper draws conclusions that do not seem well supported by the data.

To make a general point at the outset, using RMS as the sole measure of disagreement is hazardous. For example, in Figure 2, how do we know that the large RMS on the coast of China is not just one badly discrepant model rather than a Gaussian-like scatter? Or, alternatively, perhaps this RMS is large because the many global models used do not agree well: what I would want to know is how well the three most modern ones (FES2014b, TPXO9-Atlas, and GOT4.10c) agree.

Another general point is that the "East China Sea" in the title is misleading: the authors use data from the many GPS stations on Kyushu, a smaller number (but still quite a few) from the Ryukyu Islands, three in Korea, two in Taiwan, and one on the Chinese mainland. For tide gauges the same distribution is similar, except that there are six stations on the Chinese mainland and none on Korea. Any results, particularly any RMS values, will therefore be only about the first two areas, and especially Kyushu: for the GPS, the Pacific is likely to be as or more important than the East China Sea in producing almost all of the loads. I appreciate that the authors want to use as many stations as they can, but I think the paper would be much better if the few non-Japanese stations were omitted. This would also avoid a problem with Figures 4 and 5, which is that where most of the data is, it is impossible to see the results in any detail. Even if the authors do keep the few other stations, they should use a set of more focused maps, perhaps with the Kyushu-Ryukyu stations shown using an Oblique Mercator.

This geographic imbalance leads to another problem, namely the authors' conclusion that the NAO99Jb model should be used, despite its age, because of its lower RMS

compared to the tide gauges. But the authors' own Table 2 shows that for the most modern high-resolution global tide models (again, FES2014b, TPXO9-Atlas, and GOT4.10c) this lower RMS is confined to nearly-enclosed seas: for these NAO99Jb does much better. As the authors note, this is hardly surprising. The question is, how important are these enclosed seas in computing the loads?

One virtue of the station-centered grid in SPOTL is that it is very easy to combine models. So I found three polygons that enclosed these inland seas (this is very easy to do with Google Earth) and computed loads on a grid over the region. Figure 1 shows both the polygons (red) and the grid points (black): an irregular grid with smaller spacing near coastlines.

I computed loads in two ways. A was to use all of the NAO99Jb model, and TPXO7.2atlas for the remaining global parts: close to the authors' procedure. B was to use the NAO99Jb model *only* inside the polygons and TPXO7.2atlas everywhere else. Figure 2 shows the results, as contours of the ratio of the $M_2$ amplitude in vertical displacement for B, divided by the same thing for A. Two features of this plot are notable. First, the ratio is spatially smooth, which means that these enclosed seas only contribute to the estimated load for very nearby stations, so that NAO99Jb needs to be used only in these limited areas. The other is that there is, clearly, a systematic difference between loads that used NAO99Jb regionally and those that used it locally: this systematic difference might well make a difference in the authors' comparisons and conclusions. So I'd like to see the authors compute the loads using NAO99Jb only for limited areas, and more modern models (the three I've mentioned) for everywhere else.

Another major problem is that the conclusion about determining Earth structure seems inadequately supported by the evidence. Table 4 shows that once we adjust for anelastic attenuation, PREM gives RMS values that are basically indistinguishable from those for the regional model (which the authors more or less admit). Changing the model can reduce the RMS a bit more, but there is no demonstration that the reduction is significant given the added degrees of freedom: certainly the conclusion about asthenosphere depth (p. 13 lines 18-19) is not at all warranted.

A few other comments, by page and line:

8: 25-30. I have grave doubts about this method of finding errors in the loading computation. It depends, as the authors note, on the terms in the sum being uncorrelated, and that they certainly are not. So I am dubious about all subsequent invocations of errors in the loads.

In this same vein, Figure 3 shows standard deviations much larger than the RMS values of the loads from different models: this suggests that the computed errors are much too large.

I hope the final version of the paper will include a supplement with text files giving the authors' $M_2$ estimates (GPS and tide gauges) as well as the Green functions.

[Figure]

**Figure 1**

[Figure]

**Figure 2**

---

## Editor Comment (EC1) · Philip Woodworth (Editor) · 28 Oct 2019

28 October 2019

Now that the three reviews of this paper are available I thought I would add a few comments, most trivial.

The 3 reviews seem collectively very useful. Richard Ray points to the possible importance of sea water density choice in the calculations - he does not insist on it, but I wonder if it would be useful for you to at least mention it in the Conclusions, to flag the problem to readers? He also points to uncertainties with the assumed errors (Table

4 etc.) as do R2 and Duncan Agnew. R2 additionally points to a number of useful references that could be included.

The most detailed review is that of Duncan Agnew which raises questions about assumptions in the analysis to do with 'best' tide model etc. Given that this review is more detailed than the others I shall probably ask him to review your revised version.

Some detailed comments from me, many trivial:

Title etc. I though geodesists insisted on GNSS and not GPS these days? Although I can see that most of the historical record must have come from GPS.

p2, 3 - what does 'positive trend of amplitude' mean?

40 - west coast of central America

Figure 1 caption line 4 - move 'as triangles' to the end of the sentence. Also in the caption say that (b) has the same colour scale as (a) as there is no colour bar alongside it.

p6, 17-18 - what does 'instead of .. values' mean? Obviously, if you have a record of 18.6 years you need nodal corrections to be time-dependent (with that period). I guess this is alluding to some software packages for which for short records one can assume a fixed 'f' and 'u'. But for what you are doing here it is obvious they have to be at the exact times.

Figure 2 - the absence of data from S. Korea in the UHSLC data set (and also GESLA-2) is a bit of a puzzle which hopefully will be corrected at some point. It that impacts on your analysis I would be grateful if you could stress the importance.

20 - the problematic coastal areas ..

p7, 1 - is listed

Figure 3 - presumably the overflow white arrow in the colour bar is a GMT error? Could

you make that red? Also, on paper anyway, I can hardly see the three GPS station numbers in (b). Also the caption should include mention of the numbers.

23 - Ryukyu Islands respectively (Figure 3b).

p11, top - at this point I wondered if you had fully given credit to web sites or references of all data sources. Please check.

28 - I guess this 13.96 hour business is well known to GPS people but not to me. Could you have a sentence explaining more or a reference?

p12, 4 - change 'maps to only an error of' to 'has an error of only'

15 - an improvement

17 - isn't 'those' (i.e. the properties) of the asthenosphere part of the 'adopted Earth models' in the first part of the sentence? I think this needs rewording.

18 - especially important

21 'was prepared'. Sounds like cookery. You mean computed or extracted?

24 Q of 70. Is a reference needed here? Bos et al. (2015)?

p13, 27 - 'has been validated'. Is this interpretation unique?

p16, 8 needs https://

30 drop 'assumed'
* * *

---

## Author Comment (AC1) · 7 Dec 2019

Thank you for the review and comments. We provide below our responses to the points raised, including details on the modifications made to the manuscript.

*1. I want to bring up a possible systematic error which, if not addressed here, needs to be addressed at some point by this group as they continue to do these kinds of studies. I think it could cause errors of a few percent, which could be significant. Specifically, I'm beginning to think these very precise geodetic applications need to account for the variable density of seawater – that rho in Eqns (1-2) should stay inside the integral sign.*

Thank you for pointing this out. The rho (density of seawater) stays inside the integral sign in Eqns (1-2), and the latest version of SPOTL accounts for the same variable density of surface seawater as before. In order to examine the effect of the variable density of the whole column seawater, we have computed the mean seawater density for each column on a 0.25x0.25° grid using the World Ocean Atlas (Boyer et al., 2013). The result is shown in the figure below.

[Figure]

We have verified that this corresponds closely to Figure 6a of Ray et al. (2013) if $1/(g\rho_{mean})$ is plotted. For the loading computations we needed to extend some grid cells towards the coast to cover all water areas. The CARGA program was used (Bos, M S., and Baker, T. F.: An estimate of errors in gravity ocean tide loading computations, J. Geod., 79, 50-63, https://doi.org/10.1007/s00190-005-0442-5, 2005) to perform some sensitivity tests. First, a constant sea water density of 1030 kg m$^{-3}$ was used (label=1030 in Table R1.1). Second, the spatially-varying mean sea water density was used in the OTL computations (label=spatial). Finally, we corrected the mean sea water density also for compressibility using the formula given in Ray (2013), with label 'compr'. Some statistics for the changes arising in M2 height displacement amplitude and phase-lag for the 102 GPS station locations in this study are presented in Table R1.1. The mean difference for the amplitude is around 0.11 mm if compressibility is also taken into account, and this is smaller than the 0.2-0.3 mm uncertainty of the GPS observations. The maximum difference reaches 0.37 mm, which is starting to be noticeable and in future investigations this should be treated more carefully. Nevertheless, this is still much smaller than the observed discrepancies of over 1.5 mm and we have added a paragraph to the end of

Section 5 which summarises these computations and comparisons.

**Table R1.1** Influence of spatially-varying seawater density on $M_2$ vertical OTL displacement at the 102 GPS sites

| Comparison | ΔAmplitude (mm) | | | ΔPhase-lag (°) | | |
|---|---|---|---|---|---|---|
| | Min | Mean | Max | min | mean | max |
| spatial-1030 | -0.13 | 0.03 | 0.16 | -0.59 | -0.22 | 0.38 |
| compr-1030 | -0.17 | 0.10 | 0.37 | -0.99 | -0.39 | 0.65 |

*2. Abstract, line 13, recommend inserting "the regional model" before NAO99Jb, because many readers, even in the tide community, may see NAO99 and think it refers to Matsumoto's global altimeter-based model.*

We have inserted "the regional model" as suggested.

*3. Next line: "the most accurate." Well, this is risky because who knows if someone has developed another regional model here. I'd say "an accurate" – but it's up to the authors.*

We have changed the part of the Abstract to now state, "By comparison with tide gauge observations, we establish that from nine selected ocean tide models (DTU10, EOT11a, FES2014b, GOT4.10c, HAMTIDE11a, NAO99b, NAO99Jb, OSU12, TPXO9-Atlas), the regional model NAO99Jb is the most accurate in this region".

*4. In Abstract, and also page 15, bottom, authors quote 1.5 mm and 0.8 mm. I don't understand this. From Table 4, this looks to be "comparing apples and oranges." One is a maximum error and the other is RMS. Seems misleading, unless I'm just not following where they get these numbers.*

0.8 mm was not in fact the RMS but the typical maximum error arising on applying the anelastic Green's functions, which we agree should have been made clear and used statistics consistent with Table 4. We have modified Table 4 to now incorporate minimum, maximum, 90th percentile and RMS values, and we now refer to these explicitly in the Abstract, the discussion in Section 5 and in Section 6 (Conclusions).

*5. Page 13, line 5: Dahlen & Tromp is a massive book. I and many readers would appreciate your quoting the page number or even the Eqn number you're using when you cite this book.*

We have inserted the equation number (9.66) to the text where Dahlen and Tromp (1998) is cited.

*6. Page 2, line 36, where GPS is assigned an error of 0.3 mm. I don't accept this, because surely the errors in GPS are dependent on the length of the time series.*

Penna et al. (2015) showed that with 2.5 years of GPS data, a semi-diurnal harmonic displacement could be estimated with an accuracy of around 0.2-0.4 mm. This led to our choice of 0.3 mm for the GPS observational error. To clarify, we have amended the text to state "the GPS observational error is assigned a STD of 0.3 mm following Penna et al. (2015), and which assumes that at least 2.5 years of continuous GPS data will be available".

---

## Author Comment (AC2) · 7 Dec 2019

Thank you for the review and comments. We provide below our responses to the points raised, including details on the modifications made to the manuscript.

*1. The first part of the introduction gives a useful summary of previous work on using CGPS for tidal research. On page 2, line 9 it would be useful to mention that the GPS results from Yuan et al. (2013) were used by Lau, Mitrovica, ...(Nature September 2017) to look for lateral variations in body tide models of the lower mantle.*

We have added a sentence on page 2 to incorporate this.

*2. Page 3, line 16 (also page 7, line 5). It would be worth pointing out that Baker and Bos (2003, Fig. 9) used tidal gravity observations in Wuhan, China, to show that there are major problems with the earlier set of FES ocean tide models in that area.*

We have added to Section 2 the sentence, "Such problems with the earlier set of FES ocean tide models were also seen from tidal gravity observations in Wuhan, China (Baker and Bos, 2003) near this sub-area".

*3. Page 5, line 32. STDs of the phasor differences....should be STDs of the amplitudes of the phasor differences.*

We used equation 2 of Stammer et al. (2014), which we now mention in the text, and which takes the form:

$$SD = \sqrt{\frac{1}{n}\sum_{n=1}^{n}\frac{1}{2}\left[\left(H_n\cos G_n - H_{mean}\cos G_{mean}\right)^2 + \left(H_n\sin G_n - H_{mean}\sin G_{mean}\right)^2\right]}$$

$$H_{mean}\cos G_{mean} = \frac{1}{n}\sum_{n=1}^{n}H_n\cos G_n$$

$$H_{mean}\sin G_{mean} = \frac{1}{n}\sum_{n=1}^{n}H_n\sin G_n$$

*4. In Figures 4 and 5, it is very difficult to see all the phasors in Kyushu and the maps look very jumbled in that area. It is not easy to get round this problem. Maybe it would be better to reduce the number of sites/phasors and say that for clarity only 50% (or whatever) of the phasors are shown.*

We have modified both Figures 4 and 5 such that they now comprise a part (a) showing the whole region and all phasors and a part (b) showing an enlargement of Kyushu and part of the Ryukyu Islands on an oblique Mercator plot. This enables us to still show the results from all GPS sites but with less clutter.

*5. In Figure 5, the final (red) phasors still show some correlations along the Ryukyu islands. This implies that there is still some information left in these residuals. The authors may want to comment on this.*

We have included at the end of the first paragraph of the rewritten Section 5 a comment on the correlations between the residual phasors along the Ryukyu Islands, suggesting that they might be due to the tectonic setting of the subduction zone.

*6. Page 15, line 32. It is stated that the discrepancies are less than 0.8 mm on the Ryukyu Islands. This is not consistent with Table 4 on the same page.*

As per our response to Reviewer 1 point 4, we have modified Table 4 to now also include percentiles and we now refer to those in the Abstract, discussion in Section 5 and in Section 6 (Conclusions).

---

## Author Comment (AC3) · 7 Dec 2019

Thank you for the review and comments. We provide below our responses to the points raised, including details on the modifications made to the manuscript.

*1. To make a general point at the outset, using RMS as the sole measure of disagreement is hazardous. For example, in Figure 2, how do we know that the large RMS on the coast of China is not just one badly discrepant model rather than a Gaussian-like scatter? Or, alternatively, perhaps this RMS is large because the many global models used do not agree well: what I would want to know is how well the three most modern ones (FES2014b, TPXO9-Atlas, and GOT4.10c) agree.*

We use the inter-model standard deviation (as shown in Figure 2) and then in Table 2 list the RMS agreement *per model* with respect to tide gauges in each defined sub-area. For the eastern China sub-area, the per model RMS values in Table 2 indicate that there are many models contributing to the sub-area's large scatter shown in Figure 2a, and that only NAO99Jb and FES2014b are accurate there, with RMS differences of 10-12 cm, whereas TPXO9-Atlas and GOT4.10c have RMS differences of 30-35 cm. To further confirm this, in Figure R3.1 we show, per pair, plots of the phasor differences among the three most modern models, namely FES2014b-GOT4.10c, FES2014b-(TPXO9-Atlas) and GOT4.10c-(TPXO9-Atlas). It can be seen that for all three pairs very similar patterns are obtained as for the standard deviations for all nine models shown in Figure 2, and we have added a couple of sentences to Section 2 (paragraph 3) to inform the reader. The only noticeable difference is that the older model differences tend to tail off slightly less rapidly on moving from the coast. These plots, together with the RMS differences with respect to the tide gauges listed in Table 2, do not suggest that the three most modern models should be chosen over the others in this region.

[Figure]

**Figure R3.1** The $M_2$ phasor differences between each of the three most modern ocean tide models: FES2014b, GOT4.10c and TPXO9-Atlas.

*2. Another general point is that the "East China Sea" in the title is misleading: the authors use data from the many GPS stations on Kyushu, a smaller number (but still quite a few) from the Ryukyu Islands, three in Korea, two in Taiwan, and one on the Chinese mainland. For tide gauges the same distribution is similar, except that there are six stations on the Chinese mainland and none on Korea. Any results, particularly any RMS values, will therefore be only about the first two areas, and especially Kyushu: for the GPS, the Pacific is likely to be as or more important than the East China Sea in producing almost all of the loads. I appreciate that the authors want to use as many stations as they can, but I think the paper would be much better if the few non-Japanese stations were omitted. This would also avoid a problem with Figures 4 and 5, which is that where most of the data is, it is impossible to see the results in any detail. Even if the authors do keep the few other stations, they should use a set of more focused maps, perhaps with the Kyushu-Ryukyu stations shown using an oblique Mercator.*

We have changed the manuscript title to now state "around the East China Sea", not "in the East China Sea". To improve the presentation of Figures 4 and 5 (as also suggested by Reviewer 2), we now include an enlargement of Kyushu and most of the Ryukyu Islands on oblique Mercator plots. We consider the non-Japanese sites to still provide useful information as they are further from the local loads and hence provide a control on deeper mantle behaviour. Regarding the load contribution from the Pacific Ocean, we agree that a significant (but not dominant) proportion of the OTL is caused by this (as shown by Figure R3.2 below and in Table 3); but again the non-Japanese sites are useful in widening the aperture of our array to allow its effects to be distinguished from local loads. The Rest of NAO99Jb phasors in Figure R3.2 include the part of the Pacific Ocean contributing most to the loading, but inter-model variations are small, as they also are for the Central ECS contribution. The biggest impact on the loading and errors at the three sites considered comes from the very local sea areas, and where the importance of using NAO99Jb is shown.

[Figure]

**Figure R3.2** Phasor plot of the M$_2$ vertical OTL displacement contributions from the water sub-areas Eastern China, Korea, Central ECS, Seto Inland Sea (including the Kanmen Straits), Ryukyu Islands, Rest of NAO99Jb (comprises all the NAO99Jb coverage except the aforementioned sub-areas) and Rest of World.

*3. This geographic imbalance leads to another problem, namely the authors' conclusion that the NAO99Jb model should be used, despite its age, because of its lower RMS compared to the tide gauges. But the authors' own Table 2 shows that for the most modern high-resolution global tide models (again, FES2014b, TPXO9-Atlas, and GOT4.10c) this lower RMS is confined to nearly-enclosed seas: for these NAO99Jb does much better. As the authors note, this is hardly surprising. The question is, how important are these enclosed seas in computing the loads?*

*I computed loads in two ways. A was to use all of the NAO99Jb model, and TPXO7.2atlas for the remaining global parts: close to the authors' procedure. B was to use the NAO99Jb model only inside the polygons and TPXO7.2atlas everywhere else. Figure 2 shows the results, as contours of the ratio of the M2 amplitude in vertical displacement for B, divided by the same thing for A. Two features of this plot are notable. First, the ratio is spatially smooth, which means that these enclosed seas only contribute to the estimated load for very nearby stations, so that NAO99Jb needs to be used only in these limited areas. The other is that there is, clearly, a systematic difference between loads that used NAO99Jb regionally and those that used it locally: this systematic difference might well make a difference in the authors' comparisons and conclusions. So I'd like to see the authors compute the loads using NAO99Jb only for limited areas, and more modern models (the three I've mentioned) for everywhere else.*

We disagree that the lower RMS of NAO99Jb over FES2014b, GOT4.10c and TPXO9-Atlas (compared with tide gauge observations) is confined to the nearly-enclosed Ariake and Seto Inland seas: this is only the case for FES2014b. As shown in Table 2, the NAO99Jb $M_2$ RMS error with respect to tide gauges for the eastern China sub-area is 11.7 cm, whereas for GOT4.10c and TPXO9-Atlas the error is much larger at 30.1 cm and 34.5 cm respectively. Then for the Ryukyu Islands sub-area, TPXO9-Atlas has an RMS error of 11.0 cm compared with 2.4 cm for NAO99Jb, 3.1 cm for FES2014b and 3.9 cm for GOT4.10c. For the open sea areas where there are no tide gauges, Figure 2 suggests that all of FES2014b, GOT4.10c, TPXO9-Atlas and NAO99Jb agree to within 1-2 cm inter-model station deviation and so the choice of model here is immaterial.

To evaluate how important the regional improvements of NAO99Jb are in computing the load, we computed six sets of $M_2$ vertical OTL displacement for all 102 GPS sites, using two runs for each of the three most recent global models:

1. FES2014b:
    a. FES2014b augmented with all of NAO99Jb
    b. FES2014b augmented with NAO99Jb only for the Ariake and Seto Inland seas
2. GOT4.10c:
    a. GOT4.10c augmented with all of NAO99Jb
    b. GOT4.10c augmented with NAO99Jb only for the Ariake and Seto Inland seas
3. TPXO9-Atlas:
    a. TPXO9-Atlas augmented with all of NAO99Jb
    b. TPXO9-Atlas augmented with NAO99Jb only for the Ariake and Seto Inland seas

The PREM elastic Green's function was used and the minimum, maximum and RMS of the model $M_2$ vertical phasor residuals with respect to the GPS observations are shown in Table R3.1 for each ocean tide model combination. It can be seen that for all three models, the residuals when using all of NAO99Jb are smaller than when using it only for the Ariake and Seto Inland seas. The differences obtained among the global models when augmented with all of NAO99Jb are indistinguishable,

which confirms that our use of FES2014b (which was based on the accuracy tests with respect to the available tide gauges in the East China Sea region) for the Green's function comparisons is valid. This indistinguishability is also consistent with the small contributions and close model agreements for the areas outside the NAO99Jb extents shown in Figure R3.2. Whilst TPXO9-Atlas has a slightly lower RMS for the NAO99Jb augmentation with only the Ariake and Seto Inland seas compared with FES2014b and GOT4.10c, this is likely a result of improvements in areas with no tide gauges, but using all of NAO99Jb still gives lower RMS and maximum residual values.

**Table R3.1** Phasor differences (in mm) between $M_2$ vertical OTL displacement and GPS observations at the 102 GPS sites using three different global ocean tide models and different augmentations of the regional NAO99Jb model, all with the PREM elastic Green's function

| Model | (a) Use of all NAO99Jb | | | (b) NAO99Jb for Ariake and Seto Inland seas only | | |
|---|---|---|---|---|---|---|
| | Min | Max | RMS | Min | Max | RMS |
| 1. FES2014b | 0.08 | 1.59 | 0.53 | 0.06 | 1.80 | 0.69 |
| 2. GOT4.10c | 0.06 | 1.70 | 0.53 | 0.01 | 1.82 | 0.68 |
| 3. TPXO9-Atlas | 0.09 | 1.64 | 0.50 | 0.02 | 1.76 | 0.58 |

*4. Another major problem is that the conclusion about determining Earth structure seems inadequately supported by the evidence. Table 4 shows that once we adjust for anelastic attenuation, PREM gives RMS values that are basically indistinguishable from those for the regional model (which the authors more or less admit). Changing the model can reduce the RMS a bit more, but there is no demonstration that the reduction is significant given the added degrees of freedom: certainly the conclusion about asthenosphere depth (p. 13 lines 18-19) is not at all warranted.*

The paper has first (to end of Section 4) demonstrated that the systematic $M_2$ residuals of about 1.3 mm amplitude arise from deficiencies in the elastic PREM Earth model. Our original intention with Section 5 was to explain these deficiencies by first obtaining the optimal model for the region. However, as the reviewer points out (and we had already noted), differences between anelastic global PREM, anelastic regional S362ANI and our modified regional S362ANI are relatively small. We have rewritten and reordered Section 5 to reflect this, whereby we first consider if the original elastic S362ANI regional model results in reductions in the residuals over elastic global PREM, and the effect is small (RMS for the whole region and the Ryukyu Islands alone both reduced by 0.08 mm). We then describe how accounting for anelasticity at the $M_2$ frequency for both PREM (globally) and S362ANI (in the ECS region) reduces the residuals (S362ANI_M2 results in slightly smaller RMS values than PREM_M2), but still residuals at the ~0.7 mm level for the Ryukyu Islands remain. Therefore we can only test optimality of the Green's function by computing a range of Green's functions based on different asthenosphere depths and values of Q. However, we did not find any significant improvement over S362ANI_M2 on doing this, and have therefore taken care to avoid any suggestion or claim that our observations *require* any change in the asthenosphere depth from S362ANI for this region. Instead, we have described our search tests to ascertain optimality, but that they result in similar reductions as S362ANI_M2 for the ECS region, and also how, for this region, using the global PREM_M2 model leads to residuals of almost comparable size. As well as rewriting Section 5, we have modified both the Abstract and the Conclusions to reflect this. For Figure 5, we now show the S362ANI_M2 residuals, not those of mod_S362ANI_M2.

*5. I have grave doubts about this method of finding errors in the loading computation. It depends, as the authors note, on the terms in the sum being uncorrelated, and that they certainly are not. So I am dubious about all subsequent invocations of errors in the loads.*

*In this same vein, Figure 3 shows standard deviations much larger than the RMS values of the loads from different models: this suggests that the computed errors are much too large.*

The method presented is intended, to first order, to enable a further indication to be obtained that the large, systematic 1.3 mm $M_2$ residuals seen across the Ryukyu Islands and also on Kyushu are not caused by errors in the most accurate NAO99Jb model (as ascertained in Section 2). The inter-model standard deviations shown in Figure 3a for all nine ocean tide models are only around 0.3 mm across the Ryukyu Islands, suggesting that ocean tide model error contributions do not explain the discrepancies. However, around parts of Kyushu, namely the Seto Inland Sea and the Ariake Sea, the standard deviations increase to 2.5 mm, but in these regions we showed in Section 2 that the global models are erroneous. In terms of evaluating the contributions of the errors in each of the sub-area polygons to the total error at a specific location, in practice correlations are likely to exist among the polygons. For neighbouring regions, these correlations are likely to be positive, so equation 4 provides a very conservative (upper) bound on the expected level of model error. We have modified the text of Section 3 to emphasise this. For the Ryukyu Islands, the errors derived from equation 4 are about 0.5 mm, still much smaller than the 1.3 mm GPS observational residuals, and similarly across Kyushu, they are around 0.3 mm, indicating that the 2.5 mm inter-model standard deviations shown in Figure 3a are caused by models other than NAO99Jb. The errors within our defined eastern China sub-area polygon are around 1-2 mm and larger than the inter-model standard deviations, but this is because we have defined our eastern China polygon as considerably larger than the discrepant inter-model areas very close to the coast for both the ocean tides (Figure 2a) and OTL displacements (Figure 3a). Hence a conservative NAO99Jb error of 11.7 cm across all of the eastern China polygon has been applied, but which has been computed using eight tide gauges within the polygon, most of which are within the near coastal zone of much larger inter-model agreement than arises for much of the eastern China polygon. As well as mentioning in Section 3 that these large errors arise because of the fairly large 11.7 cm RMS error for NAO99Jb used, we have added an explanation that this is likely very conservative and results in errors which are too large for much of the area.

*6. I hope the final version of the paper will include a supplement with text files giving the authors' M2 estimates (GPS and tide gauges) as well as the Green functions.*

The GPS-estimated total vertical $M_2$ amplitudes and Greenwich phase lags and $M_2$ tide gauge estimates are included as a text file supplement. The Green's functions have also been moved from the appendices to this text file.

---

## Author Comment (AC4) · 7 Dec 2019

Thank you for the editorial summary and the additional comments on the manuscript itself. Our responses to each point raised, together with details of modifications made, are included below.

*Title etc. I though geodesists insisted on GNSS and not GPS these days? Although I can see that most of the historical record must have come from GPS.*
As we have used only GPS (not multi-GNSS) data in the processing and analysis, we consider GPS to be the appropriate and accurate term.

*p2, 3 - what does 'positive trend of amplitude' mean?*
We have rephrased this sentence to now read, "Ito et al. (2009) found the average amplitude ratios between GPS tidal displacement observations and an Earth tidal model (including OTL and Earth body tide) across Japan were greater than one, indicating observational agreement with inelastic Earth models".

*40 - west coast of central America*
We have inserted 'western' to clarify.

*Figure 1 caption line 4 - move 'as triangles' to the end of the sentence. Also in the caption say that (b) has the same colour scale as (a) as there is no colour bar alongside it.*
We have made these modifications.

*p6, 17-18 - what does 'instead of .. values' mean? Obviously, if you have a record of 18.6 years you need nodal corrections to be time-dependent (with that period). I guess this is alluding to some software packages for which for short records one can assume a fixed 'f' and 'u'. But for what you are doing here it is obvious they have to be at the exact times.*
We have simplified the sentence to now just state, "For time series shorter than 18.6 years, we applied nodal corrections during the harmonic tidal analysis (Foreman et al., 2009)."

*Figure 2 - the absence of data from S. Korea in the UHSLC data set (and also GESLA-2) is a bit of a puzzle which hopefully will be corrected at some point. It that impacts on your analysis I would be grateful if you could stress the importance.*
We have added a sentence to Section 2 to state that unfortunately no data are currently available within the Korea sub-area. However, it can be seen from Table 3 that the contribution of this water sub-area to the loading displacement at our sample sites shown is very small (<0.2 mm).

*20 - the problematic coastal areas ..*
'coastal' inserted.

*p7, 1 - is listed*
"are" changed to "is".

*Figure 3 - presumably the overflow white arrow in the colour bar is a GMT error? Could you make that red? Also, on paper anyway, I can hardly see the three GPS station numbers in (b). Also the caption should include mention of the numbers.*
We have modified the colour bar in Figure 3 to be red, and also the caption to now mention the three GPS stations shown. We have also improved the readability of the three GPS station names on the figure by adding a white background box.

*23 - Ryukyu Islands respectively (Figure 3b).*
Change made.

*p11, top - at this point I wondered if you had fully given credit to web sites or references of all data sources. Please check.*
Here we state the data set / networks used. They have been mentioned in full in the Data Availability section, and also the providers thanked in the Acknowledgements section.

*28 - I guess this 13.96 hour business is well known to GPS people but not to me. Could you have a sentence explaining more or a reference?*
We introduce the concept of using an artificial harmonic displacement to quality control the GPS estimation of tidal displacement at the start of Section 4, but appreciate that we had not linked the last sentence of paragraph 1 of Section 4.2 sufficiently. We have therefore amended the last two sentences of this paragraph so that they now read, "In each daily solution, an artificial 13.96 hour harmonic signal of 3.0 mm amplitude was introduced in each of the east, north and vertical components, with the phase referenced to zero defined at GPS time frame epoch J2000, and hence the GPS harmonic estimation capability with the aforementioned GIPSY processing settings assessed. 13.96 hours was chosen as the period of this displacement following Penna et al. (2015), as it is approximately in the semi-diurnal band but is distinct from the main tidal harmonics so will not be contaminated by geophysical signals."

*p12, 4 - change 'maps to only an error of' to 'has an error of only'*
We consider the original form to be a clearer description, so have left this unchanged.

*15 - an improvement*
We have changed "the improvement" to "an improvement".

*17 - isn't 'those' (i.e. the properties) of the asthenosphere part of the 'adopted Earth models' in the first part of the sentence? I think this needs rewording.*

First sentence of Section 5 now changed to read, "As Green's functions essentially depend on the material properties of the adopted Earth models, an improvement of the agreement between GPS-observed and predicted OTL values (reduction in the observational residuals) could be expected by modifying the Earth models, and the representation of the asthenosphere has been demonstrated to be especially important (Bos et al., 2015)."

*18 - especially important*

Change made (incorporated within the response to the line 17 suggestion).

*21 'was prepared'. Sounds like cookery. You mean computed or extracted?*

Section 5 has been completely rewritten in response to .Reviewer 3's comments.

*24 Q of 70. Is a reference needed here? Bos et al. (2015)?*

Kustowski et al. (2008) reference added.

*p13, 27 - 'has been validated'. Is this interpretation unique?*

We have re-written Section 5 and no longer include a statement about asthenophere validation.

*p16, 8 needs https://*

We have added http://.

*30 drop 'assumed'*

This description has been moved into Section 5 and the 'assumed' dropped.

---

## Author Response (AR2)

Thank you for the additional comments on the manuscript. Our responses to each point raised, together with details of modifications made, are included below.

**Additional comments from Editor:**

*p12, 23 - I still think this would read better as:*
*Since the NAO99Jb ocean tide has an error of only 0.2-0.5 ...*
We disagree, as it is not the NAO99Jb ocean tide model itself that has an error of 0.2-0.5 mm, but we are referring to the errors in the predicted OTL displacements caused by errors in the NAO99Jb model. Therefore we have retained the original form.

*p14,36 - et al.,*
Comma added.

*p17 - along with the Data Availability statement and the other things, I think there should be a paragraph here mentioning the Supplementary Material (see examples in other papers in this journal).*
*Also there is no reference in the paper as far as I can see to Tables S1 and S2. The PREM Green's function is said to be 'in the supplement' on p8,21 but I think this would better read 'to be in the Supplementary Material S3.' (but see the comment on S3 by Reviewer 3 below). There is no way the reader can find these unless they are all mentioned in the paper itself.*
Within the manuscript, we now explicitly reference the four tables in the Supplement, and have added a 'Supplement' paragraph on p17, pointing to a to-be-determined DOI.

**Additional comments from R3 (Duncan Agnew):**

*I remain sceptical of the author's claim that NAO99_jB is "better"—at least over most of the area it covers. But my comments on this are available to interested readers, so I will not belabor this point.*
We respect this difference of opinion.

*I think that in the local maps in Figures 4 and 5 the arrow scale could be made 50% larger, maybe even twice as large, with some gain in seeing amplitudes, without too much clutter from overlapping.*
We have enlarged the arrow scale by 50%.

*On line 2 of page 15 "on inputting" should be "using."*
Change made.

*I very much like Table 4: this set of statistics is much more informative than just the RMS.*

Thanks for the comment.

*I think the community would be better served if the S3 were split into a pdf describing what is in a separate .txt file that gave the actual numbers for the Green functions. These numbers can be extracted from the .pdf but it takes some work.*

Thank you for this valuable suggestion. We have converted and split the S3 Green's function supplement file to two ASCII .txt files.

With regard to R3's previous comment about the title, as described on our previous rebuttal we had accordingly changed the title within the manuscript to "Asthenospheric anelasticity effects on ocean tide loading *around* the East China Sea  observed with GPS" (italics and strikethrough to highlight change). However, the WWW submission interface did/does not allow us to make this change to the overall title. We ask the Editor and handling staff to ensure that the change is made throughout the final version.

[revised manuscript text omitted]